# Brittle Fracture Behavior of Sn-Ag-Cu Solder Joints with Ni-Less Surface Finish via Laser-Assisted Bonding

**DOI:** 10.3390/ma17143619

**Published:** 2024-07-22

**Authors:** Seonghui Han, Sang-Eun Han, Tae-Young Lee, Deok-Gon Han, Young-Bae Park, Sehoon Yoo

**Affiliations:** 1Regional Industry Innovation Department (Growth Engine), Korea Institute of Industrial Technology, Incheon 21999, Republic of Korea; han6755@kitech.re.kr (S.H.); sangeun35@kitech.re.kr (S.-E.H.); 2School of Materials Science and Engineering, Andong National University, Andong 36729, Republic of Korea; 3School of Materials Science and Engineering, Sungkyunkwan University, Suwon 16419, Republic of Korea; 4School of Materials Science and Engineering, Tech University of Korea, Siheung 15073, Republic of Korea; lty1226@tukorea.ac.kr; 5MK Chem & Tech Co., Ltd., Ansan 15434, Republic of Korea; deokgon.han@gmail.com

**Keywords:** laser-assisted bonding (LAB), direct electroless gold (DEG), brittle fracture characteristics, solder joint, intermetallic compound

## Abstract

In this study, we investigated the brittle fracture behavior of Sn-3.0Ag-0.5Cu (SAC305) solder joints with a Direct Electroless Gold (DEG) surface finish, formed using laser-assisted bonding (LAB) and mass reflow (MR) techniques. Commercial SAC305 solder balls were used to ensure consistency. LAB increases void fractions and coarsens the primary β-Sn phase with higher laser power, resulting in a larger eutectic network area fraction. In contrast, MR produces solder joints with minimal voids and a thicker intermetallic compound (IMC) layer. LAB-formed joints exhibit higher high-speed shear strength and lower brittle fracture rates compared to MR. The key factor in the reduced brittle fracture in LAB joints is the thinner IMC layer at the joint interface. This study highlights the potential of LAB in enhancing the mechanical reliability of solder joints in advanced electronic packaging applications.

## 1. Introduction

To enhance the performance, power, area, and cost (PPAC) of semiconductors used in AI and high-performance computing, it is crucial to increase the I/O density in semiconductor package interconnects, which requires a finer bump pitch [1]. Correspondingly, to accommodate the finer bump pitch, the pad pitch of the substrate must also be miniaturized. The conventional Electroless Nickel–Electroless Palladium–Immersion Gold (ENEPIG) surface finish can present problems due to the thick 5 μm Ni plating layer, making it unsuitable for fine-pitch applications as it cannot adequately support miniaturized pad pitches [2]. Additionally, the ferromagnetic nature of Ni causes signal loss in high-frequency 5G communication. Therefore, Ni-less surface finishes like Direct Electroless Gold (DEG) and Direct Palladium–Immersion Gold (DPIG) are being developed to eliminate the Ni plating layer [3,4].

Among electronic packaging interconnection methods, mass reflow (MR) boasts the advantage of high productivity. However, the extended process time means that the chip and substrate are exposed to high temperatures for prolonged periods. This can lead to unwanted damage to components and cause warpage due to accumulated thermal stress [5]. Warpage, in particular, can result in defects such as solder bridging and bump tearing in package interconnections, significantly reducing reliability [1,6]. As the pitch of the bumps has become finer, the amount of solder in the bumps has been reduced to mitigate solder bridging. Consequently, warpage can cause phenomena such as open defect, where bumps with a small amount of solder fail to properly hold the chip and the substrate, leading to bump detachment. Therefore, minimizing warpage is crucial as the pitch of the bumps becomes finer. Recently, laser-assisted bonding (LAB) has gained attention as a technology capable of reducing warpage compared to MR [7,8,9]. The LAB method allows for bonding by applying temperature only to a localized, desired area, such as the chip. This localized heating means that areas other than the chip and its immediate surroundings remain at a low temperature, thereby reducing warpage. Additionally, LAB offers a faster processing time compared to MR. Jang et al. [10] measured the warpage of a flip chip package manufactured using both MR and LAB through thermomechanical analysis and actual measurements. Their findings confirmed that LAB resulted in about 1/3 of the warpage compared to MR, demonstrating that LAB is an effective interconnect method for suppressing warpage.

Research on the mechanical properties and joint interface of solder joints using the LAB method has been actively conducted recently [11,12,13,14,15]. The LAB method can quickly reach the desired temperature, and the cooling rate is faster than that of MR. This allows the IMC at the solder joint interface to be more thin [16]. In general, the thicker the IMC, the more likely brittle fracture occurs [17,18]. Therefore, solder joints formed with LAB are expected to have less brittle fracture compared to the MR method, which creates a thicker IMC layer. However, there is still limited research on the brittle fracture characteristics of solder joints using the LAB method.

This study distinguishes itself by specifically investigating the brittle fracture behavior of Ni-less DEG/SAC305 solder joints formed by LAB, a topic that has received limited attention in the existing literature. Such SAC305 solder offers several advantages, including excellent mechanical properties, high thermal fatigue resistance, and superior wettability, and new bonding technologies such as nano-solders, transient liquid phase bonding, and formic acid fluxless bonding, as well as laser soldering, are being developed to enhance electronic packaging [19,20,21,22,23]. Unlike previous studies that mainly focus on mechanical properties or warpage reduction, this research explores the microstructural aspects and the influence of varying laser powers on fracture characteristics. Additionally, this study provides a comprehensive comparison between the LAB and mass reflow (MR) techniques, offering new insights into the reliability and performance enhancements achievable with LAB in advanced electronic packaging applications. By applying LAB, this research aims to understand the relationship between the interfacial microstructure of DEG/SAC305 solder joints and their brittle fracture characteristics, thereby contributing novel insights that can inform future advancements in semiconductor packaging technologies.

## 2. Materials and Methods

The substrate used in this study is shown in Figure 1. As illustrated, a flame-retardant-4 (FR-4) printed circuit board (PCB) with a solder mask-defined (SMD) copper pad was used. Figure 2 presents a schematic diagram of the surface finishes applied in this study. DEG, a Ni-less surface finish, was used as the surface finish. The pad opening size of the substrate was 350 µm. The thickness of the Au plating layer in DEG surface finish was 0.15 µm. For comparison, the brittle fracture rate of the conventional ENEPIG surface finish was also evaluated. In ENEPIG, the Ni thickness was 5 µm, the Pd thickness was 0.15 µm, and the Au thickness was 0.15 µm.

To mount solder balls on a surface-finished Cu pad, SAC305 paste (M705-GRN360-K2-V, Senju Metal, Tokyo, Japan) was applied to the DEG-finished Cu pad using a screen printer (MK-878SV, Minami, Japan). SAC305 solder balls (ET20135P 450, Duksan Hi-Metal, Ulsan, Republic of Korea) with a size of 450 µm were then placed on the printed solder paste. The SAC305 solder balls on the DEG surface finish were bonded using the LAB technique. Figure 3 shows a schematic diagram of the LAB equipment and the area covered by the laser beam on the substrate. The LAB equipment (INYA 1000W, INLASER, Gwacheon-si, Republic of Korea) used ytterbium fiber laser with a wavelength of 1070 ± 10 nm. The laser beam size of LAB was 15 × 15 mm, as shown in Figure 3b.

The LAB process included an activation step and a main bonding step. The activation step activated flux to remove oxide film from the metal surface, and the main bonding step was used to join the solder balls. In the activation step, the laser power was 0.53 W/mm^2^ for 2 s and the peak temperature was 165 °C. In the main bonding section, the laser powers were 1.56 W/mm^2^, 1.78 W/mm^2^, and 2.0 W/mm^2^ each for 2 s. The peak temperatures for LAB were 250 °C, 262 °C, and 272 °C at 1.56, 1.78, and 2.0 W/mm^2^, respectively. For comparison, the MR process was also conducted, with bonding performed at a peak temperature of 260 °C for 6 min using a reflow oven (1809UL N2, Heller, Troy, MI, USA). After bonding the solder balls with LAB, the sample image was observed with an optical microscope (OM) and is shown in Figure 4.

After the joining process, X-ray equipment (XSCAN-H160-OCT, XAVIS, Seongnam-si, Republic of Korea) was used to observe the void percentage. To evaluate the mechanical properties and brittle fracture characteristics of the solder joint, we used a high-speed shear tester (Dage 4000HS, Nordson, Westlake, OH, USA). The conditions for the high-speed shear test were a shear speed of 1 m/s and a shear height of 50 µm from the substrate surface. Based on these conditions, the calculated strain rate was 20,000 s^−1^. We tested 25 samples for each condition and classified them into three fracture modes, ductile, mixed, and brittle, based on the fracture surface analysis. The classification was performed by evaluating the percentage of the brittle area on the fracture surface. If the brittle area was 0–25%, it was classified as ductile; if it was 25–75%, it was classified as mixed; and if it was 75–100%, it was classified as brittle.

The internal microstructure of the solder ball and the microstructure of the IMC layer at the DEG/SAC305 solder joint interface were then observed using a scanning electron microscope (SEM, Inspect F, FEI, Hillsboro, OR, USA). The specimen was mounted with epoxy resin to observe the cross-section of the solder joint. After epoxy mounting, the specimen was polished using SiC paper and alumina suspensions of 3, 1, 0.3, and 0.05 µm. The chemical composition of the fracture surface and the joint interface was analyzed using energy-dispersive X-ray spectroscopy (EDS, Superdry II, Thermo Fisher Scientific, Waltham, MA, USA). The thickness of the IMC layer at the solder joint interface was measured using image analyzing software (Image J v1.53e, National Institutes of Health, Bethesda, MD, USA), and the average IMC thickness was calculated by comparing the area of the IMC layer to the interface length.

## 3. Results

### 3.1. Void Observation after LAB Process

Figure 5 shows an X-ray image taken after bonding the solder ball to the board. In the sample for the X-ray observation, only one row of solder balls was mounted, as shown in Figure 5. It was observed that under MR conditions and at a laser power of 1.56 W/mm^2^, almost no voids were present. As the laser power increased, the number of voids also increased. In contrast, solder balls formed using the MR process had almost no voids. A quantitative evaluation of the void fraction showed a value of 1.4% at a laser power of 1.78 W/mm^2^. Increasing the laser power to 2.0 W/mm^2^ increased the void fraction to 1.9%. According to Sweatman et al. [24], the void fraction in solder balls is influenced by the amount of volatiles produced from the flux and the time it takes for these volatiles to escape from the molten solder. If there is enough time for the solder to remain molten, the void fraction decreases because the volatiles from the flux residue have enough time to escape. However, if the melting time is short, the void fraction increases as there is insufficient time for the volatiles to escape. In the LAB process, where the melting time was very short, the void fraction tended to be higher compared to that in the MR process. Additionally, as the laser power increased, which increased the temperature in the LAB process, the amount of volatile generation increased, leading to a higher void fraction. In the case of MR, the void fraction was very small because there was sufficient time for the voids to escape.

### 3.2. Microstructure

The microstructure of Sn-Ag-Cu solder balls was observed using SEM according to different LAB laser power levels, as shown in Figure 6. Sn-3.0Ag-0.5Cu solder formed a eutectic network around the primary β-Sn phase. Figure 7 is an enlarged SEM/EDS view of the eutectic network. The eutectic network was composed of numerous IMC particles. IMC particle #1 in Figure 7 was identified as Ag_3_Sn, with an atomic percentage ratio of Ag to Sn of approximately 3:1, and IMC particle #2 was identified as Cu_6_Sn_5_, with an atomic percentage ratio of Cu to Sn of approximately 6:5. Therefore, it was confirmed that the eutectic network consists of Ag_3_Sn and Cu_6_Sn_5_ particles within a Sn matrix. The typical microstructure of SAC305 is primarily composed of a tin-rich matrix with dispersed intermetallic compounds, such as Ag_3_Sn and Cu_6_Sn_5_. Such eutectic network formation in an SAC305 solder alloy is generally due to the consistent composition of commercial solder balls that promote the eutectic microstructure. As the laser power of the LAB increased, the size of the β-Sn phase also increased, and the area of the eutectic network phase became larger. When the laser power went up, the temperature rose, which caused the microstructure to become coarser. Additionally, the microstructure created by LAB was finer than that of MR (mass reflow). In the MR method, the β-Sn phase became coarser because the specimen was exposed to high temperatures for a longer time compared to the LAB method.

Figure 8 shows an SEM image of the DEG/SAC305 solder joint interface. At this interface, Cu_6_Sn_5_ IMC was observed under all joint method conditions. While DEG provides a protective gold layer, this layer is typically very thin (150 nm in this study). During the soldering process, the thin gold layer dissolves quickly into the molten solder, exposing the underlying copper. This allows the tin in the solder to react with the copper, forming Cu_6_Sn_5_ IMCs. This process is similar to what occurs in an Organic Solderability Preservative (OSP) surface finish with SAC305 solder, where Cu_6_Sn_5_ IMC also forms due to a direct Sn-Cu interaction [25,26]. It was found that the IMC layer in the LAB method was much thinner compared to the MR method. Under LAB conditions, the thickest IMC layer was observed at the highest laser power of 2.0 W/mm^2^. To quantitatively measure the thickness of the IMC layer, the area of the IMC layer and the length of the joint interface were obtained using image analysis software. The average thickness was then calculated using the following equation:Average Thickness = Area of IMC Layer/Length of Joint Interface(1)
the average IMC thicknesses are shown in Figure 9. The IMC thickness for the MR method was 3.16 μm, the thickest among all bonding conditions. The IMC thickness increased with higher laser power in the LAB method: 0.69 μm at 1.56 W/mm^2^, 1.14 μm at 1.78 W/mm^2^, and 1.5 μm at 2.0 W/mm^2^.

### 3.3. Joint Strength and Brittle Fracture Rate

To evaluate the joint strength and brittle fracture behavior, a high-speed shear test was performed on solder balls joined using the LAB and the MR. Figure 10 shows the results of the high-speed shear strength test. For the high-speed shear strength test, a total of 25 specimens were observed to calculate the average value, and through a *t*-test, it was confirmed that each shear strength value was statistically different. The average joint strength of the joint using the MR method was 106 MPa, and the average joint strengths of the joint using the LAB method were 124, 119, and 115 MPa under the conditions of 1.56, 1.78, and 2.0 W/mm^2^, respectively. It was observed that all bonding conditions using the LAB method had higher bonding strengths compared to the MR method. The highest bonding strength was achieved under the condition of 1.56 W/mm^2^. As the laser power increased, the bonding strength tended to decrease, but the difference was not significant.

After the high-speed shear test, a fracture surface was observed using an SEM, as shown in Figure 11. The percentage of brittle fracture was calculated by converting the area of brittle fracture to a percentage of the total fracture surface. The fractures were then classified into three modes based on this percentage:Ductile fracture mode: 0–25% brittle fracture.Mixed fracture mode: 25–75% brittle fracture.Brittle fracture mode: 75–100% brittle fracture.

Figure 12 shows schematics for each type of failure mode. Figure 12a illustrates the ductile mode, where failure occurs inside the solder. Figure 12b represents the mixed mode, showing the fracture surface when complex failure happens both inside the solder and in the IMC layer. Figure 12c depicts the brittle mode, where failure occurs in the IMC layer. Figure 13 shows the EDS results of the ductile, mixed, and brittle mode fracture surfaces and images mapping the Cu and Sn elements, respectively. In the ductile fracture mode, fracture occurred mainly inside the SAC305 solder, and because the Cu content in the solder was relatively low at 0.5 wt%, Cu appeared mostly black in the EDS mapping. The EDS analysis showed that ductile fracture occurred mostly in the solder. On the other hand, in the brittle fracture mode, a high Cu distribution was seen in the EDS mapping, and fracture occurred in the Cu_6_Sn_5_ IMC layer, with Cu at 53.07 at% and Sn at 46.93 at% in the EDS analysis. Some solder remained at the edges of the brittle fracture surface. In the mixed fracture mode, fracture occurred both inside the SAC305 solder and in the Cu_6_Sn_5_ IMC layer. In the Sn EDS mapping of the mixed mode, the dark area in the center is presumed to be due to the height difference in the fracture surface.

Figure 14 shows a graph that quantitatively illustrates the fracture modes in DEG/SAC305 solder joints, categorized by fracture mode. It was found that all conditions using LAB had a lower incidence of brittle fracture compared to MR. In the LAB 1.56 W/mm^2^ and 1.78 W/mm^2^ conditions, fractures occurred within the 0 to 25% range, thus evaluated as a 100% ductile mode. Additionally, the LAB 2.0 W/mm^2^ condition, which had the highest laser power among the LAB methods, showed a brittle fracture mode with 28% of mixed mode and 17% of brittle mode. The incidence of the brittle fracture mode tended to increase with higher laser power. The MR method exhibited a much higher brittle fracture than the LAB method, with 28% classified as brittle and 65% as mixed, leaving only 7% as ductile.

The microstructural factors influencing brittle fracture at the solder interface include the thickness of the interfacial IMC, void near the interface, and plate-shaped Ag_3_Sn phases near the interface [20,27,28,29,30]. In this study, no large Ag_3_Sn phases were observed in the microstructure of the solder, suggesting no influence from this factor. Voids are another factor affecting mechanical strength; the larger and more numerous the voids, and the closer they are to the interface, the more the mechanical properties deteriorate [31,32]. In this study, the void fraction was largest at 1.9% in the LAB 2.0 W/mm^2^ condition, indicating that the void fraction was low and did not significantly impact brittle fracture. Moreover, since the MR condition, which had almost no voids, showed an increased brittle fracture percentage, the void fraction was not considered to influence the brittle fracture rate under the various bonding conditions of this study.

The thickness of the interfacial IMC is known to be a major cause of brittle fracture. Generally, prolonged aging at high temperatures resulted in thicker IMC layers and reduced bonding strength [33]. Thicker IMC layers promoted crack initiation under mechanical stress, reducing the lifetime of solder joints because the von Mises stress required to initiate a crack decreased as the IMC layer increased [34]. In this study, the lowest-power LAB condition, with the thinnest IMC layer, showed the highest ductile fracture mode fraction, while increasing laser power led to thicker IMC layers and higher brittle fracture mode fractions. Additionally, the MR method, which formed the thickest IMC layer, showed a higher incidence of brittle modes compared to the LAB method. When comparing the brittle fracture rates of the 2.0 W/mm^2^ condition with the MR method, the MR method had an IMC layer about twice as thick, resulting in approximately twice as many brittle fractures compared to the LAB 2.0 W/mm^2^ condition. These results emphasize that the LAB method, due to its shorter soldering time, produces thinner IMC layers than MR, leading to lower brittle fracture rates. Therefore, LAB has advantages in reducing brittle fractures and improving joint reliability.

## 4. Conclusions

In this study, the LAB and MR techniques were evaluated for their effects on the microstructure and brittle fracture properties of a Ni-less DEG surface finish and SAC305 solder joints. The key findings are as follows:From X-ray observation, no voids were observed in the MR and LAB 1.56 W/mm^2^ conditions. The void ratios increased up to 1.9% with increasing laser power.The IMC thickness was 3.16 μm with MR, while LAB showed thinner IMC layers, ranging from 0.69 μm to 1.5 μm as laser power increased.The LAB method resulted in higher shear strength than MR, with the highest strength at 1.56 W/mm^2^. Shear strength slightly decreased with higher laser power.The MR method exhibited higher brittleness than LAB, and as laser power increased, the brittleness of the LAB samples increased.Increased IMC thickness was correlated with higher brittle fracture rates.

This study concluded that the LAB method is superior in minimizing brittle fractures and maintaining higher mechanical strength due to its ability to lower IMC thickness. These findings emphasize the potential of LAB for enhancing the reliability and performance of solder joints in advanced electronic packaging applications, particularly in high-density and high-performance semiconductor devices.

## Figures and Tables

**Figure 1 materials-17-03619-f001:**
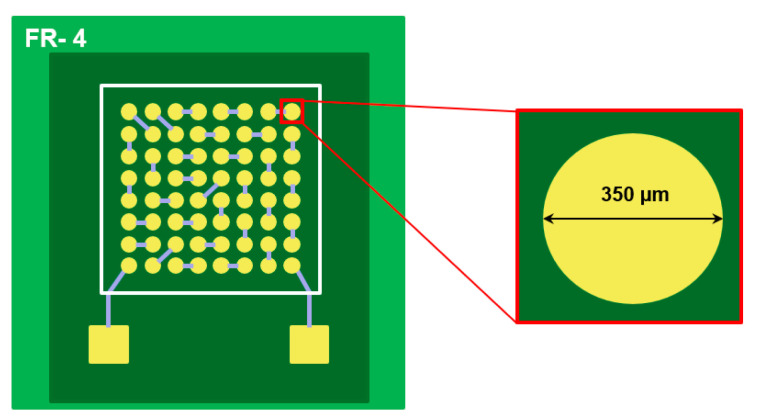
Schematics of test substrate in this study. The yellow circles represent the Cu pads, the dark green area indicates the photoimageable solder resist (PSR), and the light green area represents the substrate.

**Figure 2 materials-17-03619-f002:**
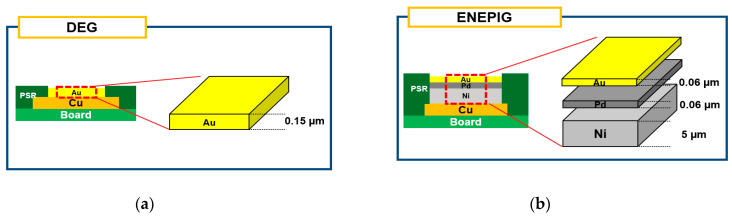
Schematic diagrams of (**a**) DEG and (**b**) ENEPIG surface finishes on Cu pads in the test substrate.

**Figure 3 materials-17-03619-f003:**
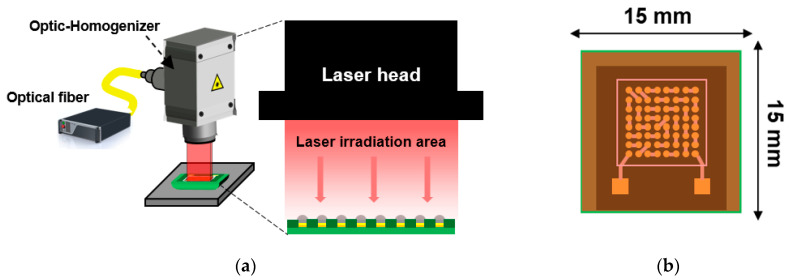
Schematic diagrams of (**a**) LAB equipment and (**b**) area covered by the laser beam on the substrate.

**Figure 4 materials-17-03619-f004:**
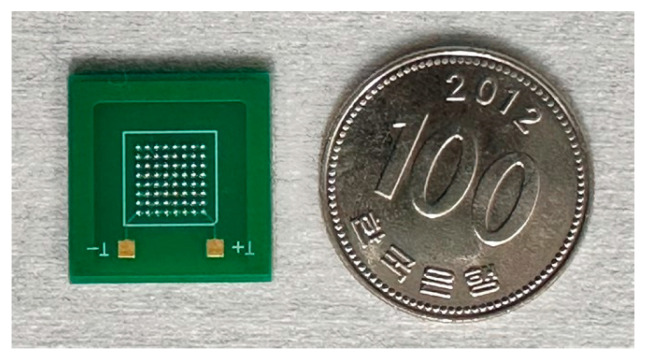
Image of solder ball bonded substrate with LAB process.

**Figure 5 materials-17-03619-f005:**
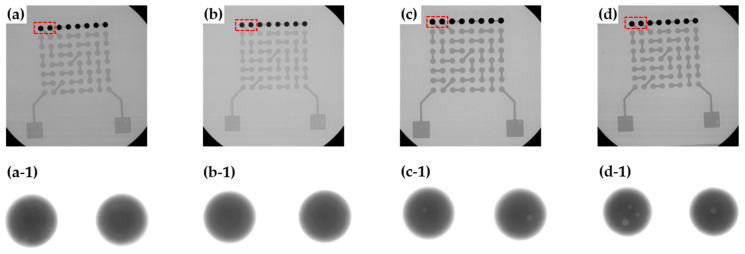
X-ray image of solder balls on test substrate: (**a**) MR, (**b**) LAB 1.56 W/mm^2^, (**c**) LAB 1.78 W/mm^2^, and (**d**) LAB 2.0 W/mm^2^. (**a-1**), (**b-1**), (**c-1**), and (**d-1**) are high magnification images of (**a**), (**b**), (**c**), and (**d**), respectively.

**Figure 6 materials-17-03619-f006:**
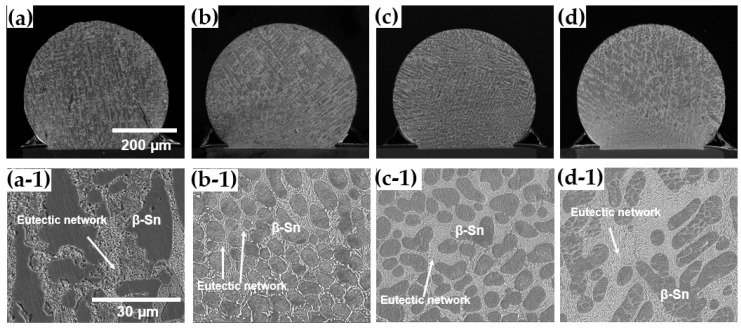
Cross-sectional SEM micrographs of SAC305 solder inside with bonding conditions: (**a**) MR, (**b**) LAB 1.56 W/mm^2^, (**c**) LAB 1.78 W/mm^2^, and (**d**) LAB 2.0 W/mm^2^. (**a-1**), (**b-1**), (**c-1**), and (**d-1**) are high magnification images of (**a**), (**b**), (**c**), and (**d**), respectively.

**Figure 7 materials-17-03619-f007:**
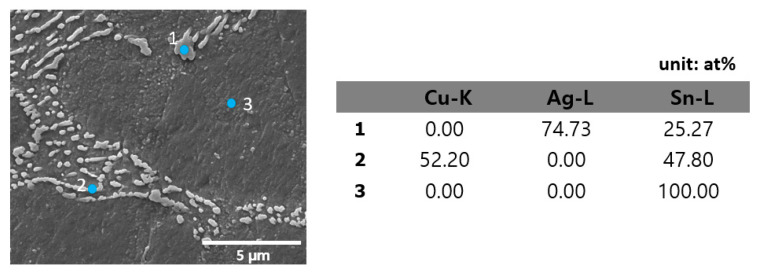
Cross-sectional SEM micrographs and EDS analysis of eutectic network in SAC305 solder.

**Figure 8 materials-17-03619-f008:**
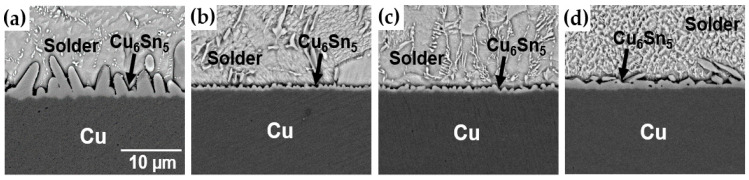
Cross-sectional SEM micrographs of joint interface of DEG/SAC305 with bonding conditions. (**a**) MR, (**b**) LAB 1.56 W/mm^2^, (**c**) LAB 1.78 W/mm^2^, and (**d**) LAB 2.0 W/mm^2^.

**Figure 9 materials-17-03619-f009:**
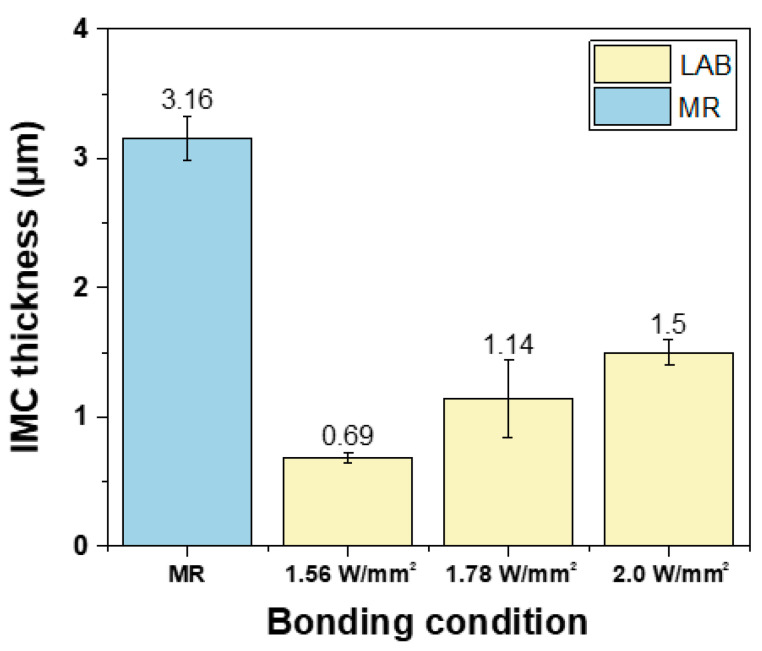
Thicknesses of interfacial IMC of DEG/SAC305 with various bonding conditions.

**Figure 10 materials-17-03619-f010:**
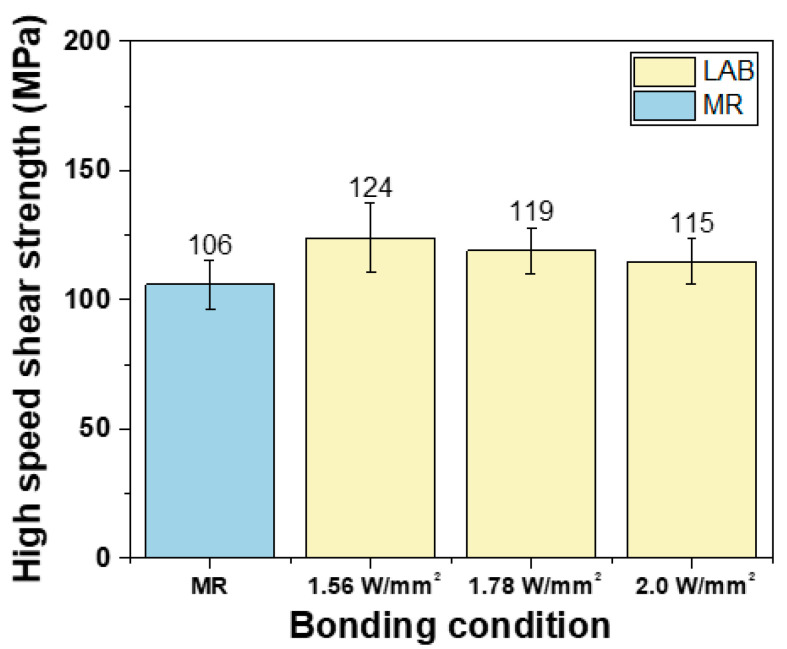
High-speed shear strengths of DEG/SAC305 with various bonding conditions.

**Figure 11 materials-17-03619-f011:**
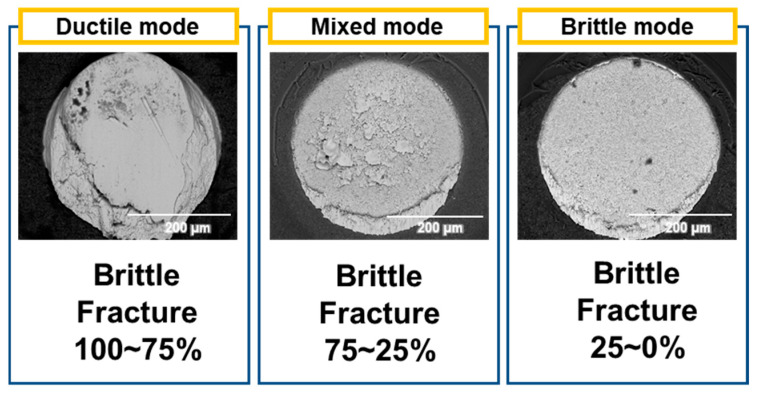
Fracture mode classification in this study.

**Figure 12 materials-17-03619-f012:**
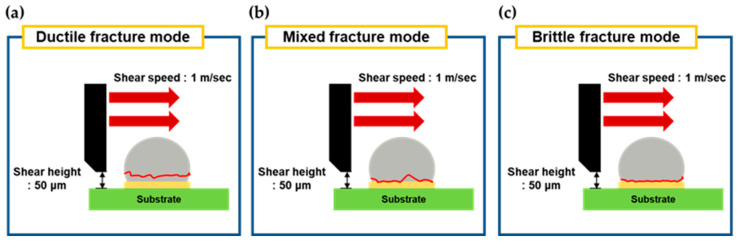
Schematic diagrams of 3 fracture modes. (**a**) ductile mode, (**b**) mixed mode, and (**c**) brittle mode.

**Figure 13 materials-17-03619-f013:**
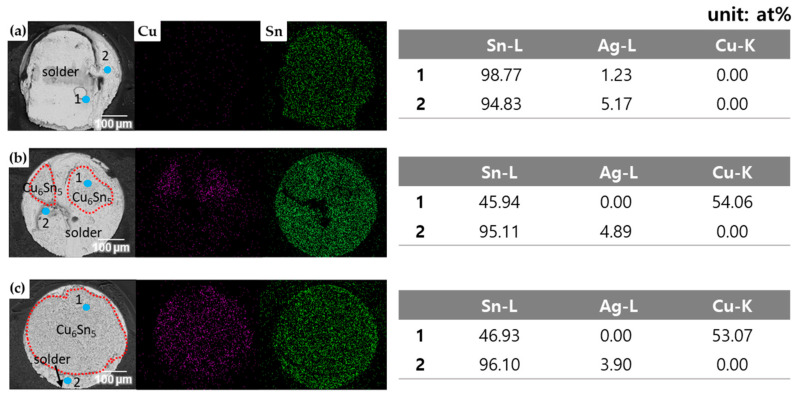
Cu and Sn EDS mapping and EDS results of (**a**) ductile fracture mode, (**b**) mixed fracture mode, and (**c**) brittle fracture mode.

**Figure 14 materials-17-03619-f014:**
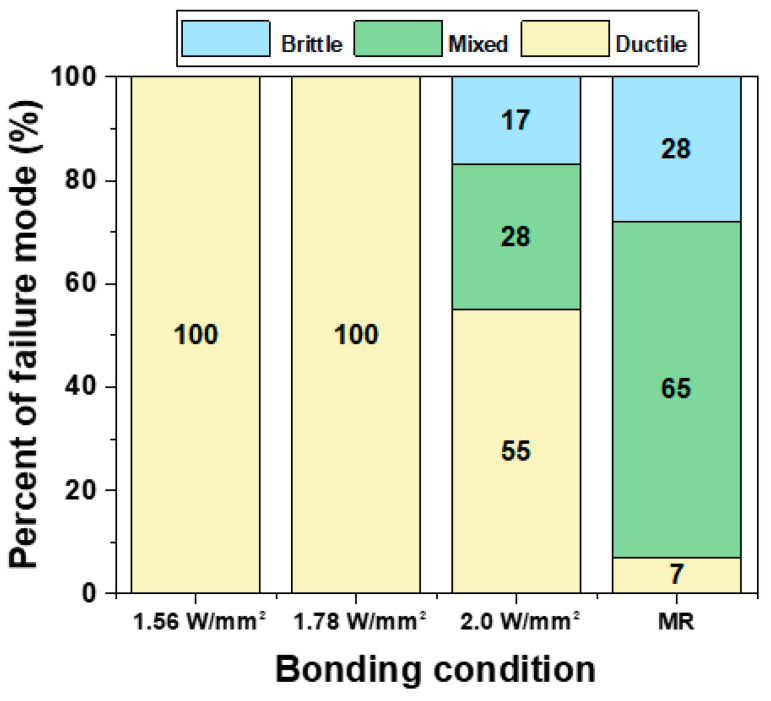
Percentage of fracture modes of DEG/SAC305 with bonding conditions.

## Data Availability

The original contributions presented in the study are included in the article, further inquiries can be directed to the corresponding authors.

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
