# Peer review of "Brittle Fracture Behavior of Sn-Ag-Cu Solder Joints with Ni-Less Surface Finish via Laser-Assisted Bonding"

_materials, 2024, doi:10.3390/ma17143619_

Round 1
Reviewer 1 Report
Comments and Suggestions for Authors
This study provides the effect and mechanism of Laser-Assisted Bonding on solder joint strength. This article presents some interesting results, and I just have a few comments for the author to consider before publishing.
1. The abbreviation for Intermetallic compound (IMC) is mentioned in the abstract, introduction, and Materials and Methods. It is recommended that the author only mention it once in the article.
2. Could the author provide the strain rate of the high-speed shear test according to the pad size?
3. According to Figure 12, SAC305 should also have Cu6Sn5 detected by EDS, but the result doesn't seem to exist. Is that correct?
4. The key to creating thin IMC layers in laser soldering is to shorten the heating time. It is suggested that the author could provide the reflow time (MR) for readers to compare with laser soldering.
5. (Optional) Authors are recommended to cite the new relevant literature to demonstrate the importance of SAC solder. Here are two articles for the author's reference.
https://doi.org/10.3390/ma17051055
https://doi.org/10.1108/SSMT-01-2024-0002
Author Response
This study provides the effect and mechanism of Laser-Assisted Bonding on solder joint strength. This article presents some interesting results, and I just have a few comments for the author to consider before publishing.
- The abbreviation for Intermetallic compound (IMC) is mentioned in the abstract, introduction, and Materials and Methods. It is recommended that the author only mention it once in the article.
Response to reviewer’s comment:
Thank you for your suggestion. We agree with the reviewer's recommendation. We have revised the manuscript accordingly to ensure that the abbreviation for "intermetallic compound (IMC)" is mentioned only once. The necessary corrections have been made in the revised manuscript as follows.
Revised section:
Line 59:
This allows the intermetallic compound (IMC) à This allows the IMC
Line 124:
The internal microstructure of the solder ball and the microstructure of the intermetallic compound (IMC) layer à The internal microstructure of the solder ball and the microstructure of the IMC layer
Line 172:
At this interface, Cu6Sn5 intermetallic compound (IMC) was observed à At this interface, Cu6Sn5 IMC was observed
- Could the author provide the strain rate of the high-speed shear test according to the pad size?
Response to reviewer’s comment:
Thank you very much for your insightful question. The shear strain rate can be calculated with following equation:
where v is shear speed and h is shear height. Here are the values provided
Shear speed v = 1 m/s =1000 mm/s
Height h = 50 μm =0.05 mm
Therefore, the strain rate = 20000 s-1
Revised section:
Line 119:
The conditions for the high-speed shear test were a shear speed of 1 m/s and a shear height of 50 µm from the substrate surface. Based on these conditions, the calculated strain rate was 20000 s⁻1
- According to Figure 12, SAC305 should also have Cu6Sn5 detected by EDS, but the result doesn't seem to exist. Is that correct?
Response to reviewer’s comment:
Thank you very much for your insightful question. In this study, EDS mapping detected the Cu element, which corresponds to the Cu in Cu6Sn5. Therefore, Cu6Sn5 is indeed present. However, as the reviewer correctly pointed out, Cu6Sn5 was not explicitly differentiated in the original figure. To address this, we have revised Figure 12 to include EDS data that explicitly distinguishes Cu6Sn5. The modified figure and corresponding explanation have been updated in the manuscript for clarity.
Revised section:
Line 227 ~ 236 :
Fig. 13 shows the EDS results of ductile, mixed, and brittle mode fracture surfaces and images mapping Cu and Sn elements, respectively. In the ductile fracture mode, fracture occurred mainly inside the SAC305 solder, and because the Cu content in the solder was relatively low at 0.5 wt%, Cu appeared mostly black in the EDS mapping. EDS analysis shows that ductile fracture occurs mostly in solder. On the other hand, in the brittle fracture mode, a high Cu distribution was seen in EDS mapping, and fracture occurred in the Cu6Sn5 IMC layer, with Cu at 53.07 at% and Sn at 46.93 at% in EDS analysis. Some solder remained at the edges of the brittle fracture surface. In the mixed fracture mode, fracture occurs both inside the SAC305 solder and in the Cu6Sn5 IMC layer. In the Sn EDS mapping of the mixed mode, the dark area in the center is presumed to be due to the height difference in the fracture surface.
Fig. 13. Cu and Sn EDS Mapping and EDS analysis results of (a) ductile fracture mode, (b) mixed fracture mode, and (c) brittle fracture mode.
- The key to creating thin IMC layers in laser soldering is to shorten the heating time. It is suggested that the author could provide the reflow time (MR) for readers to compare with laser soldering.
Response to reviewer’s comment:
Thank you very much for your suggestion. The authors have added the MR bonding process time. The added portion is highlighted in the revised manuscript.
Revised section:
Line 104~105:
For comparison, the MR process was also conducted, with bonding performed at a peak temperature of 260°C for 6 minutes using a reflow oven (1809UL N2, Heller).
- (Optional) Authors are recommended to cite the new relevant literature to demonstrate the importance of SAC solder. Here are two articles for the author's reference.
https://doi.org/10.3390/ma17051055
https://doi.org/10.1108/SSMT-01-2024-0002
Response to reviewer’s comment:
Thank you so much for your suggestion. The authors agree with the review’s recommendation. We have revised the manuscript accordingly and included citations to the new relevant literature to demonstrate the importance of SAC solder. The modified portions are highlighted in the revised manuscript.
Revised section:
Line 66-69
Such SAC305 solder offers several advantages, including excellent mechanical properties, high thermal fatigue resistance, and superior wettability, and new bonding technologies such as nano-solders, transient liquid phase bonding, and formic acid fluxless bonding as well as laser soldering are being developed to enhance electronic packaging. [19–23].
reference
- Kim, K.H.; Koike, J.; Yoon, J.W.; Yoo, S. Effect of Plasma Surface Finish on Wettability and Mechanical Properties of SAC305 Solder Joints. J. Electron. Mater. 2016, 45, 6184–6191, doi:10.1007/s11664-016-4908-4.
- Seo, W.; Kim, K.H.; Bang, J.H.; Kim, M.S.; Yoo, S. Effect of Bath Life of Ni(P) on the Brittle-Fracture Behavior of Sn-3.0Ag-0.5Cu/ENIG. J. Electron. Mater. 2014, 43, 4457–4463, doi:10.1007/s11664-014-3395-8.
- Bachok, Z.; Abas, A.; Tang, H.F.; Nazarudin, M.Z.H.; Mohd Sharif, M.F.; Che Ani, F. Investigating the Impact of Different Solder Alloy Materials during Laser Soldering Process. Solder. Surf. Mt. Technol. 2024, doi:10.1108/SSMT-01-2024-0002.
- Lin, W.; Lee, Y.C. Study of Fluxless Soldering Using Formic Acid Vapor. IEEE Trans. Adv. Packag. 1999, 22, 592–601, doi:10.1109/6040.803451.
- He, S.; Jiang, J.; Shen, Y.A.; Mo, L.; Bi, Y.; Wu, J.; Guo, C. Improvement of Solder Joint Shear Strength under Formic Acid Atmosphere at A Low Temperature. Materials (Basel). 2024, 17, 1–11, doi:10.3390/ma17051055.

Reviewer 2 Report
Comments and Suggestions for Authors
The work touches on the increasingly efficient production of electronic systems (PPAC metrics). The authors compare solder joints using two packaging methods (laser-assisted bonding and mass reflow) to develop high-quality soldering interconnections.
In general, this is not new knowledge or breakthrough discovery/research; however, specifying critical set-ups in the semiconductor industry's technological processes is a sound strategy for pushing the design limits of PPAC.
The introduction to the field of electronic device design is very decent, and the paper is also written in accessible English. However, there are a few ambiguities that I would like to draw the authors' attention to.
1. The abstract could be more straightforward and better organized. There is still enough room to improve it substantially.
2. Can the authors provide the elemental composition of the eutectic phase observed using SEM? If it were, e.g., Sn99.3Cu0.7, what would happen to the Ag excess in this solder? Would it be possible to indicate the Ag3Sn precipitates on SEM micrographs?
3. In the X-ray images presented in Figure 5, why do the mounted solder balls darken as the laser power density increases? What does this mean?
4. Comment for sentence in Lines 145-146
As far as I know, SAC305 solder alloy can form the eutectic pattern, but this is generally not typical unless one specifies remarks about commercial SAC305 solder balls used, e.g., in the semiconductor industry.
The paper is written in accessible English.
Author Response
Reviewer #2
The work touches on the increasingly efficient production of electronic systems (PPAC metrics). The authors compare solder joints using two packaging methods (laser-assisted bonding and mass reflow) to develop high-quality soldering interconnections.
In general, this is not new knowledge or breakthrough discovery/research; however, specifying critical set-ups in the semiconductor industry's technological processes is a sound strategy for pushing the design limits of PPAC.
The introduction to the field of electronic device design is very decent, and the paper is also written in accessible English. However, there are a few ambiguities that I would like to draw the authors' attention to.
1. The abstract could be more straightforward and better organized. There is still enough room to improve it substantially.
Response to reviewer’s comment:
Thank you very much for your suggestion. The authors agree with the reviewer's recommendation. We have revised the abstract to make it more straightforward and organized.
Revised section:
Line 15~24
Abstract: In this study, we investigated the brittle fracture behavior of Sn-3.0Ag-0.5Cu (SAC305) solder joints with Direct Electroless Gold (DEG) surface finish, formed using Laser-Assisted Bonding (LAB) and Mass Reflow (MR) techniques. Commercial SAC305 solder balls were used to ensure consistency. LAB increases void fractions and coarsens the primary β-Sn phase with higher laser power, resulting in a larger eutectic network area fraction. In contrast, MR produces solder joints with minimal voids and a thicker intermetallic compound (IMC) layer. LAB-formed joints exhibit higher high-speed shear strength and lower brittle fracture rates compared to MR. The key factor for the reduced brittle fracture in LAB joints is the thinner IMC layer at the joint interface. This study highlights the potential of LAB in enhancing the mechanical reliability of solder joints in advanced electronic packaging applications.
2. Can the authors provide the elemental composition of the eutectic phase observed using SEM? If it were, e.g., Sn99.3Cu0.7, what would happen to the Ag excess in this solder? Would it be possible to indicate the Ag3Sn precipitates on SEM micrographs?
Response to reviewer’s comment:
Thank you for your insightful comment. We observed the phases within the eutectic network using SEM and EDS. In the first SEM/EDS image, it can be seen that the IMC particle #1 is Ag3Sn, with an atomic percentage ratio of Ag:Sn is approximately 3:1. The particle indicated #2 is found to be Cu6Sn5, with an atomic percentage of Cu and Sn approximately 6:5. Therefore, it was determined that the eutectic network was composed of Ag3Sn and Cu6Sn5 particles within a Sn matrix. Because the content of Ag3Sn and Cu6Sn5 intermetallic compounds in the eutectic network varies from region to region, we were unable to define a consistent elemental composition of Sn, Ag, and Cu. For instance, if you observe the section marked with a square in the second SEM/EDS image, you can see that Sn is 63% in some areas and 38% in other areas, indicating that the composition is clearly different for each section. We have revised the text accordingly and highlighted the modified sections.
Revised section:
Line 158-162
Fig. 7 is an enlarged SEM/EDS view of the eutectic network. The eutectic network was composed of numerous IMC particles. IMC particle #1 in Figure 7 was identified as Ag3Sn, with an atomic percentage ratio of Ag to Sn of approximately 3:1, and IMC particle #2 was identified as Cu6Sn5, with an atomic percentage ratio of Cu to Sn of approximately 6:5. Therefore, it was confirmed that the eutectic network consists of Ag3Sn and Cu6Sn5 particles within a Sn matrix.
3. In the X-ray images presented in Figure 5, why do the mounted solder balls darken as the laser power density increases? What does this mean?
Response to reviewer’s comment:
Thank you so much for your question. In the X-ray images presented in Fig. 5, the darkening of the mounted solder balls as the laser power density increases is simply due to differences in image contrast. The contrast in the X-ray images does not change with varying the specimen. We have fixed the contrast of Fig. 5.
Revised section:
Fig. 5
Fig. 5. X-ray image of solder balls on test substrate: (a) MR, (b) LAB 1.56 W/mm², (c) LAB 1.78 W/mm², and (d) LAB 2.0 W/mm².
4. Comment for sentence in Lines 145-146
As far as I know, SAC305 solder alloy can form the eutectic pattern, but this is generally not typical unless one specifies remarks about commercial SAC305 solder balls used, e.g., in the semiconductor industry.
Response to reviewer’s comment:
Thank you for your valuable comment. The authors agree with the reviewer's comment. The typical microstructure of SAC305 is primarily composed of a tin-rich matrix with dispersed intermetallic compounds, such as Ag3Sn and Cu6Sn5. The formation of a eutectic pattern in SAC305 solder alloy is generally due to consistent composition of commercial solder balls that promote the eutectic microstructure. We have revised the manuscript to clarify this point.
Revised section:
Line 163-166
The typical microstructure of SAC305 is primarily composed of a tin-rich matrix with dispersed intermetallic compounds, such as Ag3Sn and Cu6Sn5. Such eutectic network formation in SAC305 solder alloy is generally due to consistent composition of commercial solder balls that promote the eutectic microstructure.

Reviewer 3 Report
Comments and Suggestions for Authors
the paper studies the influence of the laser power density on solder joints' mechanical and structural status.
The paper is easy to read and nicely prepared.
I only have this comment: The introduction should also point out the novelty of this paper compared to other literature.
Author Response
the paper studies the influence of the laser power density on solder joints' mechanical and structural status.
The paper is easy to read and nicely prepared.
I only have this comment: The introduction should also point out the novelty of this paper compared to other literature.
Response to reviewer’s comment:
Thank you for your positive feedback and valuable suggestion. We agree that highlighting the novelty of our study in the introduction is important. We have revised the introduction to emphasize how our research stands out compared to existing literature. The revised portion of the introduction is highlighted in the manuscript.
Revised section:
Line 64 -77
This study distinguishes itself by specifically investigating the brittle fracture behavior of Ni-less DEG/SAC305 solder joints formed by LAB, a topic that has received limited attention in existing literature. Such SAC305 solder offers several advantages, including excellent mechanical properties, high thermal fatigue resistance, and superior wettability, and new bonding technologies such as nano-solders, transient liquid phase bonding, and formic acid fluxless bonding as well as laser soldering are being developed to enhance electronic packaging [19–23]. Unlike previous studies that mainly focus on the mechanical properties or warpage reduction, this research explores the microstructural aspects and the influence of varying laser power on fracture characteristics. Additionally, this study provides a comprehensive com-parison between LAB and Mass Reflow (MR) techniques, offering new insights into the reliability and performance enhancements achievable with LAB in advanced electronic packaging applications. By applying LAB, this research aims to understand the relationship between the interfacial microstructure of DEG/SAC305 solder joints and their brittle fracture characteristics, thereby contributing novel insights that can inform future advancements in semiconductor packaging technologies.

Reviewer 4 Report
Comments and Suggestions for Authors
Dear Authors, please find my review regarding "Brittle Fracture Behavior of Sn-Ag-Cu Solder Joints with Ni-less Surface Finish via Laser Assisted Bonding "
C1: Literature survey is not comprehensive. I would like to see more regarding LAB, while the paper tackles this aspect the most. 11-15 is cited, but the overall critical thinking and discussion about the previously published material is not present. This is needed to show the overall novelty of the paper. So i'd recommend large extension of this discussion, and maybe inclusion of other articles as well.
C2: Warpage is also mentioned, however the yield/overall productivity is not connected against the advantages of LAB (like avoiding warpage). Also, how about shrinkage? Suggested ref: https://doi.org/10.1108/SSMT-07-2014-0014 How about other failure mechanisms?
C3: Materials and Methods is really well written. Maybe the image of fig4 could be contrast compensated while the background/PCB mount is confusing (also size could be highlighted with a coin, etc...). Cross-section process is missing from experimental description.
C4: X-ray contrast is not consequential along Figure 5 subimages. Please correct.
C5: Please provide a more extended statistical analysis between examples in Fig 9.
C6: What is the background of the differences of Figure 13? Why these changes occur? Please elaborate, as in depth discussion is missing.
Author Response
Dear Authors, please find my review regarding "Brittle Fracture Behavior of Sn-Ag-Cu Solder Joints with Ni-less Surface Finish via Laser Assisted Bonding "
C1: Literature survey is not comprehensive. I would like to see more regarding LAB, while the paper tackles this aspect the most. 11-15 is cited, but the overall critical thinking and discussion about the previously published material is not present. This is needed to show the overall novelty of the paper. So i'd recommend large extension of this discussion, and maybe inclusion of other articles as well.
Response to reviewer’s comment:
Thank you for your valuable feedback. We appreciate the reviewer's comments. However, we must note that there is a limited amount of research on Laser-Assisted Bonding (LAB) using area laser sources. Due to this lack of relevant papers, it is challenging to conduct a comprehensive literature survey and engage in detailed critical thinking and discussion as suggested. We have cited the available relevant references (11-15) and will continue to monitor emerging studies in this area. We believe that our work contributes significantly to this nascent field by addressing the current gaps in the literature.
C2: Warpage is also mentioned, however the yield/overall productivity is not connected against the advantages of LAB (like avoiding warpage). Also, how about shrinkage? Suggested ref: https://doi.org/10.1108/SSMT-07-2014-0014 How about other failure mechanisms?
Response to reviewer’s comment:
Thank you for your valuable feedback. The authors agree with the reviewer's comments. While it is well understood that LAB, by applying thermal energy locally and for very short durations, inherently reduces warpage and shrinkage, there is currently a lack of published literature directly linking LAB to improvements in yield and productivity. Consequently, we cannot provide a comprehensive discussion on this aspect at this time. Regarding other failure mechanisms, we are actively monitoring developments in this area and plan to address these issues in our future publications.
C3: Materials and Methods is really well written. Maybe the image of fig4 could be contrast compensated while the background/PCB mount is confusing (also size could be highlighted with a coin, etc...). Cross-section process is missing from experimental description.
Response to reviewer’s comment:
Thank you so much for your suggestion. In fact, the light green edge in Fig. 4 is not the background but a part of the PCB. What appears as a background is probably because the color of the PCB is different for each section. We have taken a new sample photo with a coin and revised Fig. 4 accordingly. Additionally, the cross-section method was added to the experimental methods section as follows.
Revised section:
Line 126~128:
The specimen was mounted with epoxy resin to observe the cross-section of the solder joint. After
epoxy mounting, the specimen was polished using SiC paper and alumina suspensions of 3, 1, 0.3, and 0.05 µm.
Fig. 4. Image of solder ball bonded substrate with LAB process.
C4: X-ray contrast is not consequential along Figure 5 subimages. Please correct.
Response to reviewer’s comment:
Thank you so much for your suggestion. We have adjusted the contrast and revised the subimages in Figure 5 accordingly.
Revised section:
Fig.5
Fig. 5. X-ray image of solder balls on test substrate: (a) MR, (b) LAB 1.56 W/mm², (c) LAB 1.78 W/mm², and (d) LAB 2.0 W/mm².
C5: Please provide a more extended statistical analysis between examples in Fig 10 (Fig. 9 à Fig.10).
Response to reviewer’s comment:
Thank you very much for your suggestion. The authors have accepted the reviewer's suggestion. Through a T-test, we confirmed that there was a difference in high-speed shear strength values according to MR and LAB conditions, and we have added the average and standard deviation values to the text.
Revised section:
Line 203-207
To evaluate the joint strength and brittle fracture behavior, a high-speed shear test was performed on solder balls joined using the LAB and the MR. Fig. 10 shows the results of the high-speed shear strength test. For the high-speed shear strength test, a total of 25 specimens were observed to calculate the average value, and through a T-test, it was confirmed that each shear strength value was statistically different. The average joint strength of the joint using the MR method was 106 MPa, and the average joint strength of the joint using the LAB method was 124, 119, and 115 MPa under the conditions of 1.56, 1.78, and 2.0 W/mm², respectively. It was observed that all bonding conditions using the LAB method had higher bonding strengths compared to the MR method. The highest bonding strength was achieved under the condition of 1.56 W/mm². As the laser power increased, the bonding strength tended to decrease, but the difference was not significant.
C6: What is the background of the differences of Figure 14 (Fig. 13àFig.14)? Why these changes occur? Please elaborate, as in depth discussion is missing.
Response to reviewer’s comment:
We really appreciate your question. As discussed in lines 260 to 283, brittle fracture can be affected by IMC thickness, voids, microstructure, etc., but in this study, IMC was judged to have a large influence. In general, the larger the interfacial IMC thickness, the higher the brittle failure rate. In Figure 9, the IMC thickness for MR reached approximately 3.16 μm, which is much thicker than the LAB specimen. In the LAB bonded sample, 2.0W/mm² had an IMC thickness of 1.5 μm, which was the highest among LAB samples. Therefore, in these samples, fracture during high-speed shear was more likely to occur at the interfacial IMC, thereby increasing the percentage of brittle mode. Figures 9 and 14 clearly show that as the thickness of the IMC layer increases, the incidence of brittle fracture also increases.

Round 2
Reviewer 1 Report
Comments and Suggestions for Authors
Thanks for the author's response. This manuscript could be accepted for publication.
Reviewer 4 Report
Comments and Suggestions for Authors
Dear Authors, thank you for improving the paper along the requested points. I see you did a proper effort in enhancing the manuscript!